# ITERATIVE TARGET AUGMENTATION FOR EFFECTIVE CONDITIONAL GENERATION

## ABSTRACT

Many challenging prediction problems, from molecular optimization to program synthesis, involve creating complex structured objects as outputs. However, available training data may not be sufficient for a generative model to learn all possible complex transformations. By leveraging the idea that evaluation is easier than generation, we show how a simple, broadly applicable, iterative target augmentation scheme can be surprisingly effective in guiding the training and use of such models. Our scheme views the generative model as a prior distribution, and employs a separately trained filter as the likelihood. In each augmentation step, we filter the model's outputs to obtain additional prediction targets for the next training epoch. Our method is applicable in the supervised as well as semi-supervised settings. We demonstrate that our approach yields significant gains over strong baselines both in molecular optimization and program synthesis. In particular, our augmented model outperforms the previous state-of-the-art in molecular optimization by over 10% in absolute gain.

## 1 INTRODUCTION

Deep architectures are becoming increasingly adept at generating complex objects such as images, text, molecules, or programs. Many useful generation problems can be seen as translation tasks, where the goal is to take a source (precursor) object such as a molecule and turn it into a target satisfying given design characteristics. Indeed, molecular optimization of this kind is a key step in drug development, though the adoption of automated tools remains limited due to accuracy concerns. We propose here a simple, broadly applicable meta-algorithm to improve translation quality.

Translation is a challenging task for many reasons. Objects are complex and the available training data pairs do not fully exemplify the intricate ways in which valid targets can be created from the precursors. Moreover, precursors provided at test time may differ substantially from those available during training — a scenario common in drug development. While data augmentation and semi-supervised methods have been used to address some of these challenges, the focus has been on either simple prediction tasks (e.g., classification) or augmenting data primarily on the source side. We show, in contrast, that iteratively augmenting translation targets significantly improves performance on complex generation tasks in which each precursor corresponds to multiple possible outputs.

Our iterative target augmentation approach builds on the idea that it is easier to evaluate candidate objects than to generate them. Thus a learned predictor of target object quality (a filter) can be used to effectively guide the generation process. To this end, we construct an external filter and apply it to the complex generative model's sampled translations of training set precursors. Candidate translations that pass the filter criteria become part of the training data for the next training epoch. The translation model is therefore iteratively guided to generate candidates that pass the filter. The generative model can be viewed as an adaptively tuned prior distribution over complex objects, with the filter as the likelihood. For this reason, it is helpful to apply the filter at test time as well, or to use the approach transductively[1] to adapt the generation process to novel test cases. The approach is reminiscent of self-training or reranking approaches employed with some success for parsing (McClosky et al., 2006; Charniak et al., 2016). However, in our case, it is the candidate generator that is complex while the filter is relatively simple and remains fixed during the iterative process.

---

[1] Allowing the model to access test set precursors (but not targets) during training.

We demonstrate that our meta-algorithm is quite effective and consistent in its ability to improve translation quality in the supervised setting. On a program synthesis task (Bunel et al., 2018), under the same neural architecture, our augmented model outperforms their MLE baseline by 8% and their RL model by 3% in top-1 generalization accuracy (in absolute measure). On molecular optimization (Jin et al., 2019a), their sequence to sequence translation baseline, when combined with our target data augmentation, achieves a new state-of-the-art result and outperforms their graph based approach by over 10% in success rate. Their graph based methods are also improved by iterative target augmentation with more than 10% absolute gain. The results reflect the difficulty of generation in comparison to evaluation; indeed, the gains persist even if the filter quality is reduced somewhat. Source side augmentation with unlabeled precursors (the semi-supervised setting) can further improve results, but only when combined with the filter in the target data augmentation framework. We provide ablation experiments to empirically highlight the effect of our method and also offer some theoretical insights for why it is effective.

## 2 Related Work

**Molecular Optimization** The goal of molecular optimization is to learn to modify compounds so as to improve their chemical properties. Jaques et al. (2017); You et al. (2018); Popova et al. (2018) used reinforcement learning approaches, while Jin et al. (2019a;b) formulated this problem as graph-to-graph translation and significantly outperformed previous methods. However, their performance remains imperfect due to the limited size of given training sets. Our work uses property prediction models to check whether generated molecules have desired chemical properties. Recent advances in graph convolutional networks (Duvenaud et al., 2015; Gilmer et al., 2017) have provided effective solutions to predict those properties *in silico*. In this work, we use an off-the-shelf property prediction model (Yang et al., 2019) to filter proposed translation pairs during data augmentation.

**Program Synthesis** Program synthesis is the task of generating a program (using domain-specific language) based on given input-output specifications (Bunel et al., 2018; Gulwani, 2011; Devlin et al., 2017). One can check a generated program's correctness by simply executing it on each input and verifying its output. Indeed, Zhang et al. (2018); Chen et al. (2019) leverage this idea in their respective decoding procedures, while also using structural constraints on valid programs.

**Semi-supervised Learning** Our method is related to various approaches in semi-supervised learning. In image and text classification, data augmentation and label guessing (Berthelot et al., 2019; Xie et al., 2019) are commonly applied to obtain artificial labels for unlabeled data. In machine translation, Norouzi et al. (2016) sample new targets from a stationary distribution in order to match the model distribution to the exponentiated payoff distribution centered at a single target sentence. Back-translation (Sennrich et al., 2015; Edunov et al., 2018) creates extra translation pairs by using a backward translation system to translate unlabeled sentences from a target language into a source language. In contrast, our method works in the forward direction because many translation tasks are not symmetric. Moreover, our data augmentation is carried out over multiple iterations, in which we use the augmented model to generate new data for the next iteration.

In syntactic parsing, our method is closely related to self-training (McClosky et al., 2006). They generate new parse trees from unlabeled sentences by applying an existing parser followed by a reranker, and then treat the resulting parse trees as new training targets. However, their method is not iterative, and their reranker is explicitly trained to operate over the top $k$ outputs of the parser; in contrast, our filter is independent of the generative model. In addition we show that our approach, which can be viewed as iteratively combining reranking and self-training, is theoretically motivated and can improve the performance of highly complex neural models in multiple domains. Co-training (Blum & Mitchell, 1998) and tri-training (Zhou & Li, 2005; Charniak et al., 2016) also augment a parsing dataset by adding targets on which multiple baseline models agree. Instead of using multiple learners, our method uses task-specific constraints to select correct outputs.

## 3 Iterative Target Augmentation

Our iterative target augmentation framework can be applied to any conditional generation task with task-specific constraints. For example, molecular optimization (Jin et al., 2019a;b) is the task of transforming a given molecule $X$ into another compound $Y$ with improved chemical properties,

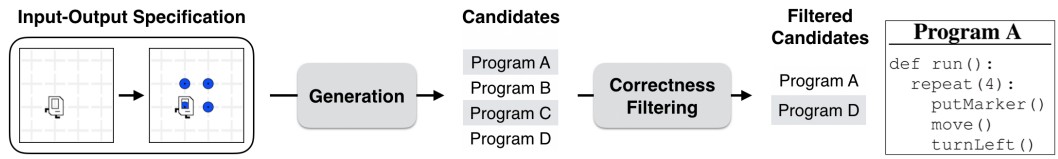

Figure 1: Illustration of our data generation process in the program synthesis setting. Given an input-output specification, we first use our generation model to generate candidate programs, and then select correct programs using our external filter. Images of input-output specification and the program A are from Bunel et al. (2018).

---

**Algorithm 1** Augmentation by iterative target augmentation

**Input:** Original training set $\mathcal{D} = [(X_1, Y_1), \ldots, (X_n, Y_n)]$

1: **procedure** AUGMENTDATASET($\mathcal{D}, \mathcal{M}_t$)
2:     $\mathcal{D}_{t+1} = \mathcal{D}$                                              ▷ Initialize augmented dataset.
3:     **for** $(X_i, Y_i)$ in $\mathcal{D}$ **do**
4:         **for** attempt in $1, \ldots, C$ **do**
5:             Apply model $\mathcal{M}_t$ to $X_i$ to sample candidate $Y'$
6:             **if** $Y'$ passes external filter **then**
7:                 Add $(X_i, Y')$ to $\mathcal{D}_{t+1}$
8:             **if** $K$ successful translations added **then**
9:                 break from loop
10:     **return** augmented dataset $\mathcal{D}_{t+1}$

11: **procedure** TRAIN($\mathcal{D}$)
12:     **for** epoch in $1, \ldots, n_1$ **do**                                    ▷ Regular training
13:         Train model on $\mathcal{D}$.
14:     **for** epoch in $1, \ldots, n_2$ **do**                                    ▷ Iterative target augmentation
15:         $\mathcal{D}_{t+1} = $ AUGMENTDATASET($\mathcal{D}, \mathcal{M}_t$)
16:         $\mathcal{M}_{t+1} \leftarrow$ Train model $\mathcal{M}_t$ on $\mathcal{D}_{t+1}$.

---

while constraining $Y$ to remain similar to $X$. Program synthesis (Bunel et al., 2018; Chen et al., 2019) is the task of generating a program $Y$ satisfying input specification $X$; for example, $X$ may be a set of input-output test cases which $Y$ must pass.

Without loss of generality, we formulate the generation task as a translation problem. For a given input $X$, the model learns to generate an output $Y$ satisfying the constraint $c$. The proposed augmentation framework can be applied to any translation model $\mathcal{M}$ trained on an existing dataset $\mathcal{D} = \{(X_i, Y_i)\}$. As illustrated in Figure 1, our method is an iterative procedure in which each iteration consists of the following two steps:

- **Augmentation Step**: Let $\mathcal{D}_t$ be the training set at iteration $t$. To construct each next training set $\mathcal{D}_{t+1}$, we feed each input $X_i \in \mathcal{D}$ (the original training set, not $\mathcal{D}_t$) into the translation model up to $C$ times to sample $C$ candidate translations $Y_i^1 \ldots Y_i^C$.[2] We take the first $K$ distinct translations for each $X_i$ satisfying the constraint $c$ and add them to $\mathcal{D}_{t+1}$. When we do not find $K$ distinct valid translations, we simply add the original translation $Y_i$ to $\mathcal{D}_{t+1}$.

- **Training Step**: We continue to train the model $\mathcal{M}_t$ over the new training set $\mathcal{D}_{t+1}$ for one epoch.

The above training procedure is summarized in Algorithm 1. As the constraint $c$ is known a priori, we can construct an external filter to remove generated outputs that violate $c$ during the augmentation step. At test time, we also use this filter to screen predicted outputs. To propose the final translation of a given input $X$, we have the model generate up to $L$ outputs until we find one satisfying the constraint $c$. If all $L$ attempts fail for a particular input, we just output the first of the failed attempts.

---

[2]One could augment $\mathcal{D}_t$ instead of $\mathcal{D}$ and continuously expand the dataset, but the empirical effect is small (see Appendix B.3). We note this augmentation step can be trivially parallelized if speed is a concern.

Finally, as an additional improvement, we observe that the augmentation step can be carried out for unlabeled inputs $X$ that have no corresponding $Y$. Thus we can further augment our training dataset in the transductive setting by including test set inputs during the augmentation step, or in the semi-supervised setting by simply including unlabeled inputs.

## 4   MOTIVATION FOR ITERATIVE TARGET AUGMENTATION

We provide here some theoretical motivation for our iterative target augmentation framework. For simplicity, we consider an external filter $c_{X,Y}$ that is a binary indicator function representing whether output $Y$ satisfies the desired constraint in relation to input $X$. In other words, we would like to generate $Y$ such that $Y \in B(X) = \{Y'|c_{X,Y'} = 1\}$. If the initial translation model $P^{(0)}(Y|X)$ serves as a reasonable prior distribution over outputs, we could simply "invert" the filter and use

$$P^{(*)}(Y|X) \propto P^{(0)}(Y|X) \cdot c_{X,Y} \tag{1}$$

as the ideal translation model. While this posterior calculation is typically not feasible but could be approximated through samples, it relies heavily on the appropriateness of the prior (model prior to augmentation). Instead, we go a step further and iteratively optimize our parametrically defined prior translation model $P_\theta(Y|X)$. Note that the resulting prior can become much more concentrated around acceptable translations.

We maximize the log-likelihood that candidate translations satisfy the constraints implicitly encoded in the filter

$$\mathbb{E}_X \left[\log P_\theta(c_{X,Y} = 1 \mid X)\right] \tag{2}$$

In many cases there are multiple viable outputs for any given input $X$. The training data may provide only one (or none) of them. Therefore, we treat the output structure $Y$ as a latent variable, and expand the inner term of Eq.(2) as

$$\log P_\theta(c_{X,Y} = 1 \mid X) \quad = \quad \log \sum_Y P_\theta(Y, c_{X,Y} = 1 \mid X) \tag{3}$$

$$= \quad \log \sum_Y P(c_{X,Y} = 1 \mid Y, X) P_\theta(Y|X) \tag{4}$$

$$= \quad \log \sum_Y c_{X,Y} \cdot P_\theta(Y|X) \tag{5}$$

Since the above objective involves discrete latent variables $Y$, we propose to maximize Eq.(5) using the standard EM algorithm (Dempster et al., 1977), especially its incremental, approximate variant. The target augmentation step in our approach is a sampled version of the E-step where the posterior samples are drawn with rejection sampling guided by the filter. The number of samples $K$ controls the quality of approximation to the posterior.[3] The additional training step based on the augmented targets corresponds to a generalized M-step. More precisely, let $P_\theta^{(t)}(Y|X)$ be the current translation model after $t$ epochs of augmentation training. In epoch $t + 1$, the augmentation step first samples $C$ different candidates for each input $X$ using the old model $P^{(t)}$ parameterized by $\theta^{(t)}$, and then removes those which violate the constraint $c$, interpretable as samples from the current posterior $Q^{(t)}(Y|X) \propto P_{\theta^{(t)}}(Y|X)c_{X,Y}$. As a result, the training step maximizes the EM auxiliary objective via stochastic gradient descent:

$$J(\theta \mid \theta^{(t)}) = \mathbb{E}_X \left[\sum_Y Q^{(t)}(Y|X) \log P_\theta(Y|X)\right] \tag{6}$$

We train the model with multiple iterations and show empirically that model performance indeed keeps improving as we add more iterations. The EM approach is likely to converge to a different and better-performing translation model than the initial posterior calculation discussed above.

## 5   EXPERIMENTS

We demonstrate the broad applicability of iterative target augmentation by applying it to two tasks of different domains: molecular optimization and program synthesis.

---

[3]See Appendix B.3 for details and experiments on the effect of sample size $K$.

Figure 2: Illustration of molecular optimization. Molecules can be modeled as graphs, with atoms as nodes and bonds as edges. Here, the task is to train a translation model to modify a given input molecule into a target molecule with higher drug-likeness (QED) score. The constraint has two components: the output $Y$ must be highly drug-like, and must be sufficiently similar to the input $X$.

## 5.1 MOLECULAR OPTIMIZATION

The goal of molecular optimization is to learn to modify molecules so as to improve their chemical properties. As illustrated in Figure 2, this task is formulated as a graph-to-graph translation problem. Similar to machine translation, the training set is a set of molecular pairs $\{(X, Y)\}$. $X$ is the input molecule (precursor) and $Y$ is a similar molecule with improved chemical properties. Each molecule in the training set $\mathcal{D}$ is further labeled with its property score. Our method is well-suited to this task because the target molecule is not unique: each precursor molecule can be modified in many different ways to optimize its properties.

**External Filter** The constraint for this task contains two parts: 1) the chemical property of $Y$ must exceed a certain threshold $\beta$, and 2) the molecular similarity between $X$ and $Y$ must exceed a certain threshold $\delta$. The molecular similarity $\text{sim}(X, Y)$ is defined as Tanimoto similarity on Morgan fingerprints (Rogers & Hahn, 2010), which measures structural overlap between two molecules.

In real world settings, ground truth values of chemical properties are often evaluated through experimental assays, which are too expensive and time-consuming to run for iterative target augmentation. Therefore, we construct an *in silico* property predictor $F_1$ to approximate the true property evaluator $F_0$. To train this property prediction model, we use the molecules in the training set and their labeled property values. The predictor $F_1$ is parameterized as a graph convolutional network and trained using the Chemprop package (Yang et al., 2019). During data augmentation, we use $F_1$ to filter out molecules whose predicted property is under the threshold $\beta$.

### 5.1.1 EXPERIMENTAL SETUP

We follow the evaluation setup of Jin et al. (2019b) for two molecular optimization tasks:

1. **QED Optimization**: The task is to improve the drug-likeness (QED) of a given compound $X$. The similarity constraint is $\text{sim}(X, Y) \geq 0.4$ and the property constraint is $\text{QED}(Y) \geq 0.9$, with $\text{QED}(Y) \in [0, 1]$ defined by the system of Bickerton et al. (2012).

2. **DRD2 Optimization**: The task is to optimize biological activity against the dopamine type 2 receptor (DRD2). The similarity constraint is $\text{sim}(X, Y) \geq 0.4$ and the property constraint is $\text{DRD2}(Y) \geq 0.5$, where $\text{DRD2}(Y) \in [0, 1]$ is the predicted probability of biological activity given by the model from Olivecrona et al. (2017).

We treat the output of the *in silico* evaluators from Bickerton et al. (2012) and Olivecrona et al. (2017) as ground truth, and we use them only during test-time evaluation.[4]

**Evaluation Metrics.** During evaluation, we are interested both in the probability that the model will find a successful modification for a given molecule, as well as the diversity of the successful modifications when there are multiple. We translate each molecule in the test set $Z = 20$ times, resulting in candidate modifications $Y_1 \ldots Y_Z$ (not necessarily distinct). We use the following two evaluation metrics:

---

[4]Although the Chemprop model we use in our filter is quite powerful, it fails to perfectly approximate the ground truth models for both QED and DRD2. The test set RMSE between our Chemprop model and the ground truth is 0.015 on the QED task and 0.059 on DRD2, where both properties range from 0 to 1.

| Model | QED Succ. | QED Div. | DRD2 Succ. | DRD2 Div. |
|---|---|---|---|---|
| VSeq2Seq | 58.5 | 0.331 | 75.9 | 0.176 |
| *VSeq2Seq+* (Ours) | 89.0 | 0.470 | 97.2 | 0.361 |
| *VSeq2Seq+, semi-supervised* (Ours) | **95.0** | 0.471 | **99.6** | **0.408** |
| *VSeq2Seq+, transductive* (Ours) | 92.6 | 0.451 | 97.9 | 0.358 |
| HierGNN | 76.6 | 0.477 | 85.9 | 0.192 |
| *HierGNN+* (Ours) | **93.1** | **0.514** | **97.6** | **0.418** |

Table 1: Performance of different models on QED and DRD2 optimization tasks. Italicized models with + are augmented with iterative target augmentation. We emphasize that iterative target augmentation remains critical to performance in the semi-supervised and transductive settings; data augmentation without an external filter instead decreases performance.

1. *Success*: The fraction of molecules $X$ for which *any* of the outputs $Y_1 \ldots Y_Z$ meet the required similarity and property constraints (specified previously for each task). This is our main metric.
2. *Diversity*: For each molecule $X$, we measure the average Tanimoto distance (defined as $1 - \text{sim}(Y_i, Y_j)$) between pairs within the set of successfully translated compounds among $Y_1 \ldots Y_Z$. If there are one or fewer successful translations then the diversity is 0. We average this quantity across all test molecules.

**Models and Baselines.** We consider the following two model architectures from Jin et al. (2019a) to show that our augmentation scheme is not tied to specific neural architectures.

1. VSeq2Seq, a sequence-to-sequence translation model generating molecules by their SMILES string (Weininger, 1988).
2. HierGNN, a hierarchical graph-to-graph architecture that achieves state-of-the-art performance on the QED and DRD2 tasks, outperforming VSeq2Seq by a wide margin.

We apply our iterative augmentation procedure to the above two models, generating up to $K = 4$ new targets per precursor during each epoch of iterative target augmentation. Additionally, we evaluate our augmentation of VSeq2Seq in a transductive setting, as well as in a semi-supervised setting where we provide 100K additional source-side precursors from the ZINC database (Sterling & Irwin, 2015). Full hyperparameters are in Appendix A.

### 5.1.2 RESULTS

As shown in Table 1, our iterative augmentation paradigm significantly improves the performance of VSeq2Seq and HierGNN. On both datasets, the translation success rate increases by over 10% in absolute terms for both models. In fact, VSeq2Seq+, our augmentation of the simple VSeq2Seq model, outperforms the non-augmented version of HierGNN. This result strongly confirms our hypothesis about the inherent challenge of learning translation models in data sparse scenarios. Moreover, we find that adding more precursors during data augmentation further improves the VSeq2Seq model. On the QED dataset, the translation success rate improves from 89.0% to 92.6% by just adding test set molecules as precursors (VSeq2Seq+, transductive). When instead adding 100K presursors from the external ZINC database, the performance further increases to 95.0% (VSeq2Seq+, semi-supervised). We observe similar improvements for the DRD2 task as well. Beyond accuracy gain, our augmentation strategy also improves the diversity of generated molecules. For instance, on the DRD2 dataset, our approach yields 100% relative gain in terms of output diversity.

**Importance of Property Predictor** Although the property predictor used in data augmentation is different from the ground truth property evaluator used at test time, the difference in evaluators does not derail the overall training process. Here we analyze the influence of the quality of the property predictor used in data augmentation. Specifically, we rerun our experiments using less accurate predictors in the property-predicting component of our external filter. We obtain these less accurate predictors by undertraining Chemprop and decreasing its hidden dimension. For comparison, we also report results with the oracle property predictor which is the ground truth property evaluator.

As shown in Figure 3, on the DRD2 dataset, we are able to maintain strong performance despite using predictors that deviate significantly from the ground truth. This implies that our framework can potentially be applied to other properties that are harder to predict. On the QED dataset, our

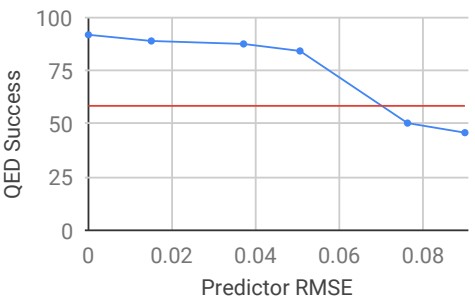 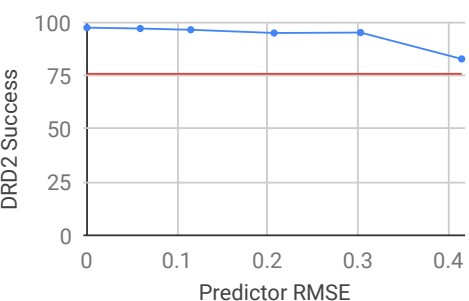

Figure 3: **Left**: QED success rate vs. Chemprop predictor's RMSE with respect to ground truth on test set. The red line shows the performance of the (unaugmented) VSeq2Seq baseline. **Right**: Same plot for DRD2. In each plot, the far left point with zero RMSE is obtained by reusing the ground truth predictor, while the second-from-left point is the Chemprop predictor we use to obtain our main results. Points further to the right are weaker predictors trained for fewer epochs and with less capacity, simulating a scenario where the property is more difficult to model.

| Model | Train | Test | QED Succ. | QED Div. | DRD2 Succ. | DRD2 Div. |
|---|---|---|---|---|---|---|
| VSeq2Seq | ✗ | ✗ | 58.5 | 0.331 | 75.9 | 0.176 |
| VSeq2Seq(test) | ✗ | ✓ | 77.4 | 0.471 | 87.2 | 0.200 |
| VSeq2Seq(train) | ✓ | ✗ | 81.8 | 0.430 | 92.2 | 0.321 |
| VSeq2Seq+ | ✓ | ✓ | **89.0** | 0.470 | **97.2** | **0.361** |
| VSeq2Seq(no-filter) | ✗ | ✗ | 47.5 | 0.297 | 51.0 | 0.185 |

Table 2: Ablation analysis of filtering at training and test time. "Train" indicates a model whose training process uses data augmentation according to our framework. "Test" indicates a model that uses the external filter at prediction time to discard candidate outputs which fail to pass the filter. The evaluation for VSeq2Seq(no-filter) is conducted after 10 augmentation epochs, as the best validation set performance only decreases over the course of training.

method is less tolerant to inaccurate property prediction because the property constraint is much tighter — it requires the QED score of an output $Y$ to be in the range $[0.9, 1.0]$.

**Importance of External Filtering** Our full model (VSeq2Seq+) uses the external filter during both training and testing. We further experiment with Vseq2seq(test), a version of our model trained without data augmentation but which uses the external filter to remove invalid outputs at test time. As shown in Table 2, VSeq2Seq(test) performs significantly worse than our full model trained under data augmentation. Similarly, a model VSeq2Seq(train) trained with the data augmentation but without the prediction time filtering also performs much worse than the full model.

In addition, we run an augmentation-only version of the model without an external filter. This model (referred to as VSeq2Seq(no-filter) in Table 2) augments the data in each epoch by simply using the first $K$ distinct candidate translations for each precursor $X$ in the training set, without using the external filter at all. In addition, we provide this model with the 100K unlabeled precursors from the semi-supervised setting. Nevertheless, we find that the performance of this model steadily declines from that of the bootstrapped starting point with each data augmentation epoch. Thus the external filter is necessary to prevent poor targets from leading the model training astray.

## 5.2 PROGRAM SYNTHESIS

In program synthesis, the source is a set of input-output specifications for the program, and the target is a program that passes all test cases. Our method is suitable for this task because the target program is not unique. Multiple programs may be consistent with the given input-output specifications. The external filter is straightforward for this task: we simply check whether the generated output passes all test cases. Note that at evaluation time, each instance contains extra held-out input-output test cases; the program must pass these in addition to the given test cases in order to be considered correct. When we perform prediction time filtering, we do not use held-out test cases in our filter.

| Model | Top-1 Generalization |
|---|---|
| MLE (Bunel et al., 2018) | 71.91 |
| MLE + RL + Beam Search (Bunel et al., 2018) | 77.12 |
| *MLE+* (Ours) | **80.17** |

Table 3: Model performance on Karel program synthesis task. MLE+ is our augmented version of the MLE model (Bunel et al., 2018).

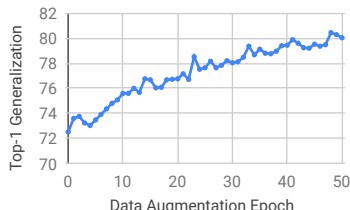

Figure 4: Top-1 generalization accuracy of MLE+ model on validation set of Karel task across different epochs.

### 5.2.1 EXPERIMENTAL SETUP

Our task is based on the educational Karel programming language (Pattis, 1981) used for evaluation in Bunel et al. (2018) and Chen et al. (2019). Commands in the Karel language guide a robot's actions in a 2D grid, and may include for loops, while loops, and conditionals. Figure 1 contains an example. We follow the experiment setup of Bunel et al. (2018).

**Evaluation Metrics.** The evaluation metric is top-1 generalization. This metric measures how often the model can generate a program that passes the input-output test cases on the test set. At test time, we use our model to generate up to $L$ candidate programs and select the first one to pass the input-output specifications (not including held-out test cases).

**Models and Baselines.** Our main baseline is the MLE baseline from Bunel et al. (2018). This model consists of a CNN encoder for the input-output grids and a LSTM decoder along with a handcoded syntax checker. It is trained to maximize the likelihood of the provided target program. Our model is the augmentation of this MLE baseline by our iterative target augmentation framework. As with molecular optimization, we generate up to $K = 4$ new targets per precursor during each augmentation step. Additionally, we compare against the best model from Bunel et al. (2018), which finetunes the same MLE architecture using an RL method with beam search to estimate gradients.[5] We use the same hyperparameters as the original MLE baseline; see Appendix A for details.

### 5.2.2 RESULTS

Table 3 shows the performance of our model in comparison to previous work. Our model (MLE+) outperforms the base MLE model in Bunel et al. (2018) model by a wide margin. Moreover, our model outperforms the best reinforcement learning model (RL + Beam Search) in Bunel et al. (2018), which was trained to directly maximize the generalization metric. This demonstrates the efficacy of our approach in the program synthesis domain. Since our augmentation framework is complementary to architectural improvements, we hypothesize that other techniques, such as execution based synthesis (Chen et al., 2019), can benefit from our approach as well.

## 6 CONCLUSION

In this work, we have presented an iterative target augmentation framework for generation tasks with multiple possible outputs. Our approach is theoretically motivated, and we demonstrate strong empirical results on both the molecular optimization and program synthesis tasks, significantly outperforming baseline models on each task. Moreover, we find that iterative target augmentation is complementary to architectural improvements, and that its effect can be quite robust to the quality of the external filter. Finally, in principle our approach is applicable to other domains as well.

---

[5]More recently, Chen et al. (2019) achieved state-of-the-art performance on the same Karel task, with top-1 generalization accuracy of 92%. They use a different architecture highly specialized for program synthesis as well as a specialized ensemble method. Thus their results are not directly comparable to our results in this paper. We did not apply our method to their model as their implementation is not publicly available.

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

# A    MODEL HYPERPARAMETERS

Our augmented models share the same hyperparameters as their baseline counterparts in all cases.

## A.1    MOLECULAR OPTIMIZATION

For the VSeq2Seq model we use batch size 64, embedding and hidden dimension 300, VAE latent dimension 30, and an LSTM with depth 1 (bidirectional in the encoder, unidirectional in the decoder). For models using iterative target augmentation, $n_1$ is set to 5 and $n_2$ is set to 10, while for the baseline models we train for 20 epochs (corresponding to $n_1 = 20, n_2 = 0$). The HierGNN model shares the same hyperparameters as in Jin et al. (2019a).

For the training time and prediction time filtering parameters, we set $K = 4, C = 200$, and $L = 10$ for both the QED and DRD2 tasks.

## A.2    PROGRAM SYNTHESIS

For the Karel program synthesis task, we use the same hyperparameters as the MLE baseline model in Bunel et al. (2018). We use a beam size of 64 at test time, the same as the MLE baseline, but simply sample programs from the decoder distribution when running iterative target augmentation during training. The baseline model is trained for 100 epochs, while for the model employing iterative target augmentation we train as normal for $n_1 = 15$ epochs followed by $n_2 = 50$ epochs of iterative target augmentation. Due to the large size of the full training dataset, in each epoch of iterative augmentation we use $\frac{1}{10}$ of the dataset, so in total we make 5 passes over the entire dataset.

For the training time and prediction time filtering parameters, we set $K = 4, C = 50$, and $L = 10$.

# B    ADDITIONAL EXPERIMENTAL DETAILS

## B.1    DATASET SIZES

In Table 4 we provide the training, validation, and test set sizes for all of our tasks. For each task we use the same splits as our baselines.

| Task | Training Set | Validation Set | Test Set |
|------|--------------|----------------|----------|
| QED | 88306 | 360 | 800 |
| DRD2 | 34404 | 500 | 1000 |
| Karel | 1116854 | 2500 | 2500 |

Table 4: Number of source-target pairs in training, validation, and test sets for each task.

## B.2    MOLECULAR OPTIMIZATION LEARNING CURVES

In Figure 5, we provide the validation set performance per iterative target augmentation epoch for our VSeq2Seq+ model on both the QED and DRD2 tasks. The corresponding figure for the MLE+ model on the Karel task is in the main text in Figure 4.

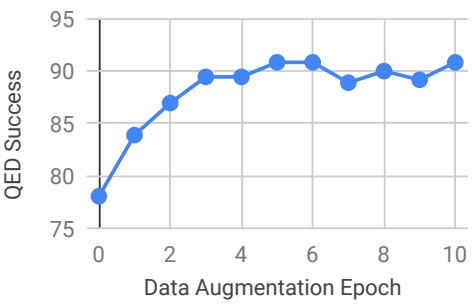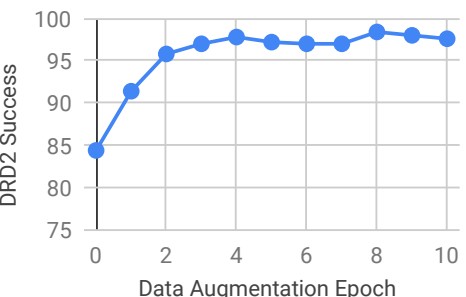

Figure 5: **Left**: QED success rate for VSeq2Seq+ on validation set for each epoch of iterative target augmentation. **Right**: Same plot for DRD2. For each plot, the far left point indicates the performance of the bootstrapped model.

### B.3 FURTHER MOLECULAR OPTIMIZATION EXPERIMENTS

In our molecular optimization tasks, we experiment with the effect of modifying $K$, the number of new targets added per precursor during each training epoch. In all other experiments we have used $K = 4$. Since taking $K = 0$ corresponds to the base non-augmented model, it is unsurprising that performance may suffer when $K$ is too small. However, as shown in Table 5, at least in molecular optimization there is relatively little change in performance for $K$ much larger than $4$.

| Model | QED Succ. | QED Div. | DRD2 Succ. | DRD2 Div. |
|---|---|---|---|---|
| *VSeq2Seq+, K=2* | 85.1 | 0.453 | 95.9 | 0.327 |
| *VSeq2Seq+, K=4* | 89.0 | 0.470 | 97.2 | 0.361 |
| *VSeq2Seq+, K=8* | 88.4 | 0.480 | 97.6 | 0.373 |

Table 5: Performance of our model VSeq2Seq+ with different values of $K$. All other experiments use $K = 4$.

We also experiment with a version of our method which continually grows the training dataset by keeping all augmented targets, instead of discarding new targets at the end of each epoch. We chose the latter version for our main experiments due to its closer alignment to our EM motivation. However, we demonstrate in Table 6 that performance gains from continually growing the dataset are small to insignificant in our molecular optimization tasks.

| Model | QED Succ. | QED Div. | DRD2 Succ. | DRD2 Div. |
|---|---|---|---|---|
| *VSeq2Seq+* | 89.0 | 0.470 | 97.2 | 0.361 |
| *VSeq2Seq+, keep-targets* | 89.8 | 0.465 | 97.6 | 0.363 |

Table 6: Performance of our proposed augmentation scheme, VSeq2Seq+, compared to an alternative version (VSeq2Seq+, keep-targets) which keeps all generated targets and continually grows the training dataset.

### B.4 PROGRAM SYNTHESIS ABLATIONS

In Table 7 we provide the same ablation analysis that we provided in the main text for molecular optimization, demonstrating that both training time iterative target augmentation as well as prediction time filtering are beneficial to model performance. However, we note that even MLE(train), our model without prediction time filtering, outperforms the best RL method from Bunel et al. (2018).

| Model | Train | Test | Top-1 Generalization |
|---|---|---|---|
| MLE* | ✗ | ✗ | 70.91 |
| *MLE(test)** | ✗ | ✓ | 74.12 |
| *MLE(train)* | ✓ | ✗ | 77.92 |
| *MLE+* | ✓ | ✓ | **80.17** |

Table 7: Ablation analysis of filtering at training and test time. "Train" indicates a model whose training process uses data augmentation according to our framework. "Test" indicates a model that uses the external filter at prediction time to discard candidate outputs which fail to pass the filter. Note that MLE and MLE(test) are based on an MLE checkpoint which underperforms the published result from Bunel et al. (2018) by 1 point, due to training for fewer epochs.

