# OpenReview forum: "Iterative Target Augmentation for Effective Conditional Generation"
_ICLR.cc/2020/Conference — Reject_

### Official Review · AnonReviewer3 · 2019-10-22
**Official Blind Review #3**

**Rating:** 6

**Review:**

This paper proposes a training scheme to enhance the optimization process where the outputs are required to meet certain constraints. The authors propose to insert an additional target augmentation phase after the regular training. For each datapoint, the algorithm samples candidate outputs until it find a valid output according the an external filter. The model is further fine-tuned on the augmented dataset. In experiments, the authors evaluated the proposed training scheme on two datasets: molecular optimization and program synthesis. Results show that the proposed training algorithm improves the success rate and the diversity among generated targets.

The improvement over the baselines is impressive and the paper seems to be well written. However, I have some concerns related to the novelty of the algorithm and some details in the experiments.

1) After reading Algorithm 1, it looks like a knowledge distillation step where the teacher model is the regular model enhanced by external filtering.

2) If you are able to train neural-based property evaluator F1, why don't just directly use F1(y) as a value network and use simple REINFORCE to create additional teacher signal for the model.

3) Another concern is that the baselines may be too weak. There may be simpler decoding strategies to guide the sequence generation with the property evaluator. For example, you can reject all invalid choices of next token in the sequence by checking the property evaluator score. A related work can be https://openreview.net/forum?id=HJIHtIJvz . Is this feasible on the datasets used in this paper?


**Experience Assessment:**

I do not know much about this area.

**Review Assessment: Checking Correctness Of Derivations And Theory:**

I did not assess the derivations or theory.

**Review Assessment: Checking Correctness Of Experiments:**

I assessed the sensibility of the experiments.

**Review Assessment: Thoroughness In Paper Reading:**

I made a quick assessment of this paper.

---

> ### Author Response · Authors · 2019-11-11
> **Response**
>
> Thank you for your insightful comments! Regarding your review:
>
> 1) After reading Algorithm 1, it looks like a knowledge distillation step where the teacher model is the regular model enhanced by external filtering.
>
> Algorithm 1 is a stochastic (and incremental) EM for the latent variable model where the predicted structure is latent and the likelihood arises from the property/similarity constraints. One could try to interpret it as a form of knowledge distillation where the teacher and the model share parameters. The teacher would presumably be the posterior distribution. We note, however, that the stochastic, incremental EM view is precisely what the algorithm does.
>
> 2) If you are able to train neural-based property evaluator F1, why don't just directly use F1(y) as a value network and use simple REINFORCE to create additional teacher signal for the model.
>
> We did compare with an RL approach in the program synthesis task. In fact, our iterative target augmentation model outperforms Bunel et al.’s more sophisticated (compared to REINFORCE) RL baseline, while using the same architecture. Their method also uses correctness of the output as a component of reward. We outperform this baseline even when omitting the prediction-time filtering component of our augmentation (Appendix B.4). Applying RL to molecular optimization is more challenging due to the sparsity of rewards. For molecules, it is problematic to predict their properties given partial sequences, because there are possibly many ground truth targets, and because not every subsequence corresponds to a valid molecule. Thus the RL reward may come only at the end of the sequence generation process. Note also that reward maximizing and log-marginal likelihood (ours) behave somewhat differently as training criteria.
>
> 3) Another concern is that the baselines may be too weak. There may be simpler decoding strategies to guide the sequence generation with the property evaluator. For example, you can reject all invalid choices of next token in the sequence by checking the property evaluator score. A related work can be https://openreview.net/forum?id=HJIHtIJvz . Is this feasible on the datasets used in this paper?
>
> We agree that Zhang et al. is an important prior work and we have added it in the related work. However, their technique is infeasible for our tasks for two reasons. First, each input can be associated with multiple unknown outputs that are all correct. As a result, we cannot reject invalid choices of next token by simply checking whether the sequence can expand into the known ground truth target. Second, the property evaluator can be a complex function such as a neural network. For molecular optimization, it is unclear how we would reject invalid choices of each next token because the property evaluator may not even work on partial sequences which do not correspond to valid molecules. In order to deal with a variety of tasks, our method assumes the property evaluator is a black-box function. A decoding strategy using the property evaluator on partial sequences would require additional, task-dependent modifications following the structure of the property evaluator.

---

### Official Review · AnonReviewer2 · 2019-10-23
**Official Blind Review #2**

**Rating:** 3

**Review:**

(post rebuttal) I appreciate the authors for detailed rebuttal. I stay with the original score based on the following reason.

In RAML you sample from exponentiated reward distribution and do maximum likelihood (minimizing forward KL in classic control as inference framework). How you sample depends on the problem assumption. In the original paper, they used a heuristic based on edit distance from ground truth, but in theirs they did not have assumption that you can evaluate correctness (as used in external filter). In this paper's context, the sampling naturally comes down to rejection sampling. Therefore I keep the original stance regarding the similarity between this work and RAML.

---

The paper proposes an iterative data augmentation approach based self-generation and filtering with success criteria. The authors justify the algorithm as an EM procedure of maximizing \log p^*(y|x), where p^*(y|x) \propto p(y|x) * p(c=1|x,y). They demonstrate that the iterative data augmentation procedure can provide significant gains to SOA models in molecule generation and outperform MLE+RL method in program synthesis datasets.

Strengths:
- A simple approach that can be added to any seq2seq translation where success metric can be evaluated efficiently
- Demonstrated results on multiple applications

Weaknesses:
- The method requires being able to evaluate constraints
- The approach is simple and does not seem to present significant novelty over prior methods. Particularly, the approach could be considered as nesting RAML [1] updates with (1) low temperature, (2) self-generated trajectories.

Other comments:
- Some missing references [1, 2]. [2] also studies molecule generation and proposes approaches similar to RL + MLE.

[1] Norouzi, Mohammad, et al. "Reward augmented maximum likelihood for neural structured prediction." Advances In Neural Information Processing Systems. 2016.
[2] Jaques, Natasha, et al. "Sequence tutor: Conservative fine-tuning of sequence generation models with kl-control." Proceedings of the 34th International Conference on Machine Learning-Volume 70. JMLR. org, 2017.

**Experience Assessment:**

I have published one or two papers in this area.

**Review Assessment: Checking Correctness Of Derivations And Theory:**

I carefully checked the derivations and theory.

**Review Assessment: Checking Correctness Of Experiments:**

I assessed the sensibility of the experiments.

**Review Assessment: Thoroughness In Paper Reading:**

I read the paper thoroughly.

---

> ### Author Response · Authors · 2019-11-11
> **Response**
>
> Thank you for your insightful comments! Regarding your review:
>
> Note that our method is not optimizing log P*(y|x) but rather log[ sum_y P(c=1|x,y) P(y|x)], i.e., log-likelihood of the "evidence" where evidence is expressed as the intersection of property constraints we want candidates y to satisfy. We train this latent variable model by introducing a stochastic version of generalized (incremental) EM.
>
> - The method requires being able to evaluate constraints
>
> The method is indeed directly guided by property predictors or constraints (the evidence). However, since regression/classification tasks are often “easier” than the corresponding generation tasks, the approach remains widely applicable to conditional generation tasks; we already demonstrated two different tasks. When a trivial rule-based evaluator (cf program synthesis) is not available, we can simply train a neural model for the associated regression/classification task. Indeed, we trained a neural property predictor for the molecular optimization task. We showed that even a relatively poor neural predictor still enables the augmented model to outperform non-augmented baselines in Section 5.1.2.
>
> - The approach is simple and does not seem to present significant novelty over prior methods. Particularly, the approach could be considered as nesting RAML [1] updates with (1) low temperature, (2) self-generated trajectories.
>
> We agree that RAML [Norouzi et al. 2016] is an important prior work, which is now included in the related work. However, beyond the implementation similarity of sampling new targets, theoretically our model (derived directly from EM) is somewhat different. RAML samples new targets from a stationary distribution in order to match the model distribution to the exponentiated payoff distribution centered at a single gold target. In contrast, there’s no straightforward way to sample directly from the set of targets satisfying our constraints (exponentiated or not), whether in molecular optimization or program synthesis. Thus our method draws samples from the possibly multimodal posterior Q(y|x) propto P(c=1|x,y) P(y|x) (stochastic EM) via rejection sampling, where the current translation model P(y|x) is used as the candidate generator. Maximizing expected reward vs log-marginal likelihood are quite different as training criteria.
>
> Moreover, nested RAML updates with low temperature and self-generated trajectories, in the absence of our external filter, corresponds to vanilla self-training; we showed in our experiments in Section 5.1.2 that this performs extremely poorly. The external filter is critical to model performance.
>
> - Some missing references [1, 2]. [2] also studies molecule generation and proposes approaches similar to RL + MLE.
>
> Thank you for the additional references, which we now include as related work.

---

### Official Review · AnonReviewer1 · 2019-10-29
**Official Blind Review #1**

**Rating:** 6

**Review:**

This paper proposes a data augmentation strategy for a class of problems that the amount of labelled data is limited while the evaluation procedure is easier. Specifically, they are able to incorporate some of the model’s output into training data to guide the training procedure. The idea is quite simple and effective according to empirical results.

Pros:
    The idea is quite simple and easy to implement. It can be applied into a broad problem. Also, it’s not tied to specific neural architecture
Section 4 provide an interpretation from view of EM algorithm, which is quite interesting.
    The improvement of empirical studies are desirable, validating the effectiveness of the proposed method on two complex tasks, including molecule optimization and program synthetic.
Cons:
    K in Algorithm 1 plays an important role. I expect more discussion of how to select K. Is K fixed during training procedure in different epochs? Is K the same magnitude as size of original training dataset.
In molecular optimization task, i suggest authors to add more details on setup. For example, on DRD2 and QED, what’s QED(X) and DRD2(X)? In success metric, what is your required constraint on both similarity and property score?
Extra:
    I don’t understand why the earlier collected data pairs are thrown away. Have you tried the strategy that incorporate all the augmented data?

**Experience Assessment:**

I have published one or two papers in this area.

**Review Assessment: Checking Correctness Of Derivations And Theory:**

I assessed the sensibility of the derivations and theory.

**Review Assessment: Checking Correctness Of Experiments:**

I assessed the sensibility of the experiments.

**Review Assessment: Thoroughness In Paper Reading:**

I read the paper at least twice and used my best judgement in assessing the paper.

---

> ### Author Response · Authors · 2019-11-11
> **Response**
>
> Thank you for your insightful comments! Regarding your review:
>
> -- K in Algorithm 1 plays an important role. I expect more discussion of how to select K. Is K fixed during training procedure in different epochs? Is K the same magnitude as size of original training dataset.
>
> K is fixed (4 in our experiments), and we have clarified that K is the number of additional targets per precursor, rather than the total number of additional targets added to the dataset. So our augmented datasets are 5 times as large as our original training dataset.
>
> Larger K correspond to improved approximations of the true posterior distribution Q^(t) (Y|X) during the E-step. So we might expect larger K to potentially improve diversity, leading to an increase in success rate, at the cost of more computation during the E-step as well as M-step (because we need to generate as well as train on more targets). Actually, we found previously that the empirical effect of K is relatively small when increasing further beyond the value K=4 that we use, at least in the molecular optimization setting. These experiments are now included in Appendix B.3. We have also provided some additional discussion and clarification on the role of K in Section 4.
>
> -- In molecular optimization task, i suggest authors to add more details on setup. For example, on DRD2 and QED, what’s QED(X) and DRD2(X)? In success metric, what is your required constraint on both similarity and property score?
>
> QED(X) is the drug-likeness score of a molecule X, which is defined by the system of Bickerton et al. (2012). DRD2(X) is the predicted probability of biological activity against the dopamine type 2 receptor, given by the model from Olivecrona et al. (2017). In the success metric, we require the similarity score >= 0.4 (the same for both tasks), QED >= 0.9 and DRD2 >= 0.5. We have clarified these details in Section 5.1.1.
>
> -- I don’t understand why the earlier collected data pairs are thrown away. Have you tried the strategy that incorporate all the augmented data?
>
> We chose to discard previously collected pairs because re-sampling new pairs at each epoch aligns more closely with the theoretical EM interpretation. Continually growing the dataset by keeping all earlier pairs is a perfectly reasonable alternative, and we did previously try this version of the model. We found that further performance gains were small to insignificant on the molecular optimization task; these experiments are now included in Appendix B.3.

---

### Decision · Program_Chairs · 2019-12-19

**Decision:**

Reject

**Comment:**

This paper proposes a training scheme to enhance the optimization process where the outputs are required to meet certain constraints. The authors propose to insert an additional target augmentation phase after the regular training. For each datapoint, the algorithm samples candidate outputs until it find a valid output according the an external filter. The model is further fine-tuned on the augmented dataset.

The authors provided detailed answers and responses to the reviews, which the reviewers appreciated. However, some significant concerns remained, and  due to a large number of stronger papers, this paper was not accepted at this time.